# UtpA and UtpB chaperone nascent pre-ribosomal RNA and U3 snoRNA to initiate eukaryotic ribosome assembly

Mirjam Hunziker[1,*], Jonas Barandun[1,*], Elisabeth Petfalski[2], Dongyan Tan[3], Clémentine Delan-Forino[2], Kelly R. Molloy[4], Kelly H. Kim[5], Hywel Dunn-Davies[2], Yi Shi[4], Malik Chaker-Margot[1,6], Brian T. Chait[4], Thomas Walz[5], David Tollervey[2] & Sebastian Klinge[1]

Early eukaryotic ribosome biogenesis involves large multi-protein complexes, which co-transcriptionally associate with pre-ribosomal RNA to form the small subunit processome. The precise mechanisms by which two of the largest multi-protein complexes—UtpA and UtpB—interact with nascent pre-ribosomal RNA are poorly understood. Here, we combined biochemical and structural biology approaches with ensembles of RNA–protein cross-linking data to elucidate the essential functions of both complexes. We show that UtpA contains a large composite RNA-binding site and captures the 5′ end of pre-ribosomal RNA. UtpB forms an extended structure that binds early pre-ribosomal intermediates in close proximity to architectural sites such as an RNA duplex formed by the 5′ ETS and U3 snoRNA as well as the 3′ boundary of the 18S rRNA. Both complexes therefore act as vital RNA chaperones to initiate eukaryotic ribosome assembly.

[1] Laboratory of Protein and Nucleic Acid Chemistry, The Rockefeller University, New York, New York 10065, USA. [2] Wellcome Trust Centre for Cell Biology, University of Edinburgh, Michael Swann Building, Max Born Crescent, Edinburgh EH9 3BF, UK. [3] Department of Cell Biology, Harvard Medical School, Boston, Massachusetts 02115, USA. [4] Laboratory of Mass Spectrometry and Gaseous Ion Chemistry, The Rockefeller University, New York, New York 10065, USA. [5] Laboratory of Molecular Electron Microscopy, The Rockefeller University, New York, New York 10065, USA. [6] Tri-Institutional Training Program in Chemical Biology, The Rockefeller University, New York, New York 10065, USA. * These authors contributed equally to this work. Correspondence and requests for materials should be addressed to S.K. (email: klinge@rockefeller.edu).

Eukaryotic ribosome assembly involves the concerted functions of >200 non-ribosomal factors, which are involved in a large range of reactions from transcriptional events in the nucleolus to final quality control steps in the cytoplasm[1]. While significant progress has been made in determining the late stages of small subunit[2,3] and large subunit maturation[4–7], relatively little is known about the early co-transcriptional events during which large ribosome assembly factors associate with nascent pre-ribosomal RNA (pre-rRNA) to form the small subunit (SSU) processome[8,9].

The SSU processome has been identified as the earliest ribosome assembly intermediate during maturation of the small ribosomal subunit. Within this complex, the nascent pre-rRNA transcript is cleaved at sites A0, A1 and A2, resulting in the release of the 20S pre-rRNA. The SSU processome is a giant particle composed of numerous ribosome assembly factors, including the UtpA, UtpB, UtpC and Mpp10 complexes, the U3 small nucleolar ribonucleoprotein (snoRNP) and many individual proteins[8,10]. We have recently analysed SSU processome formation as a function of transcription and identified a 2-MDa particle associated with the 5′-external transcribed spacer (5′ ETS). In addition to the 700-nucleotide 5′ ETS pre-rRNA, this particle contains 26 proteins, including all known components of the UtpA, UtpB, Mpp10 and U3 snoRNP complexes[11].

UtpA and UtpB are multi-protein complexes with seven and six subunits, respectively[12–14]. Elucidating their biochemical functions during ribosome synthesis has been hampered by their large size, the transient nature of pre-ribosomes and the absence of evident enzymatic activities. Both complexes predominantly contain β-propeller and α-helical repeat structures, suggesting that they form a structural framework during early ribosome assembly[3,15–18]. Depletion experiments indicate that UtpA initiates pre-ribosome assembly by binding to the nascent pre-rRNA, and then recruits UtpB and the U3 snoRNP[9,19]. The U3 snoRNP may act as an RNA chaperone by base-pairing with, and bringing together, multiple sequences in the 5′ ETS and 18S rRNA[20–22]. However, the precise pre-rRNA-binding sites and mechanisms by which UtpA and UtpB function during early stages of eukaryotic ribosome assembly in vivo have not yet been determined.

The pre-rRNA-binding sites of numerous individual ribosome assembly factors have been elucidated using UV cross-linking and analysis of cDNA (CRAC)[23–25]. With molecular weights of 660 and 525 kDa, respectively, UtpA and UtpB account for approximately half the mass of the 5′ ETS particle[11]. UtpA and UtpB have a combined number of 13 distinct subunits and are likely to have more complex and/or composite pre-rRNA-binding sites, which we determined using a systematic analysis of ensembles of RNA-protein cross-linking data from multiple subunits.

Here, we report the architecture, function and precise pre-rRNA-binding sites of these complexes using RNA–protein and protein–protein cross-linking data combined with electron microscopy (EM).

## Results

**Molecular architectures of UtpA and UtpB.** To determine the molecular structures and interactions of UtpA and UtpB, we purified both complexes from yeast using nanobodies directed against fluorescent proteins, as previously described[11,26] (Figs 1 and 2). UtpA is a heptameric complex consisting of Utp4, Utp5, Utp8, Utp9, Utp10, Utp15 and Utp17 (Nan1). The endogenous complex was purified to homogeneity via a protease-cleavable Utp10-GFP fusion protein (Fig. 1a,b) and used

for cross-linking and mass spectrometry experiments. To test the stability and salt sensitivity of UtpA, the heptameric complex was isolated under low-salt conditions (200 mM NaCl) and subsequently incubated with buffers of increasing salt concentrations. At 400 mM NaCl, Utp4 dissociated from the other six subunits (Fig. 1c) and at 800 mM NaCl only Utp17 was still bound to the immobilized Utp10-GFP (Fig. 1d). This result suggests that the large proteins Utp10 (200 kDa) and Utp17 (101 kDa) form a complex, with which the remaining five smaller subunits associate[14]. It further indicates that Utp4 is the most salt-sensitive subunit of the five smaller subunits.

To elucidate if the five smaller subunits can form a subcomplex in the absence of the Utp10-Utp17 heterodimer, Utp4, Utp5, Utp8, Utp9 and a protease-cleavable Utp15-mCherry fusion protein were overexpressed in yeast. After affinity purification, all five subunits were present (Supplementary Fig. 1), but Utp4 dissociated from Utp5, Utp8, Utp9 and Utp15-mCherry during the subsequent size-exclusion step (Fig. 1e). The loss of Utp4 in low-salt buffer conditions during size-exclusion chromatography suggests either a weak association of Utp4 with the other subunits or the necessity for Utp10 and/or Utp17 for its stable integration within UtpA.

To investigate the protein–protein interactions within UtpA further, we performed cross-linking and mass spectrometry of purified natively tagged UtpA (Fig. 1a,b and f). A dense network of cross-links was identified between Utp10, Utp5, Utp15, Utp8 and Utp17, suggesting that these are located in close proximity within UtpA. As expected, Utp10 shares cross-links with Utp17 but also shares a large number of cross-links with Utp5 and Utp8. Utp4 was not strongly cross-linked to other subunits of UtpA and only shared few cross-links with Utp8, Utp9 and Utp15 (Fig. 1f and Supplementary Fig. 2a). Taken altogether, these observations suggest a molecular organization of UtpA in which Utp10 and Utp17 form a stable dimer that has spatial proximity to Utp5, Utp8 and Utp15 and to a lesser extent to Utp9 and the salt-labile Utp4 (Fig. 1d–f).

UtpB is a hexameric complex consisting of Utp1 (Pwp2), Utp6, Utp12 (Dip2), Utp13, Utp18 and Utp21. The endogenous complex was purified using a protease-cleavable Utp21-mCherry fusion protein and anti-mCherry nanobody-covered sepharose resin[26]. Affinity purification and protease cleavage followed by size-exclusion chromatography confirms the presence of all six subunits previously reported to be part of UtpB[13] (Fig. 2a,b), which was further confirmed by mass spectrometry (data not shown).

We set out to reconstitute subcomplexes of UtpB in bacteria to identify which subunits interact directly with each other within UtpB. Several subcomplexes were obtained by heterologous co-expression and purification of UtpB subunits from bacteria (Fig. 2c–f). Super-stoichiometric amounts of tagged Utp6 were observed in some experiments (Fig. 2d,f), indicating the recovery of monomeric tagged protein in addition to the stoichiometric complex. Utp12 and Utp13 co-elute on size-exclusion chromatography, and thus interact directly to form a stable heterodimeric complex (Fig. 2c, Supplementary Fig. 3a,b). Similarly, the co-expression of Utp6 and Utp18 yields a stable heterodimer (Fig. 2d). These two heterodimeric subcomplexes of UtpB are bridged via Utp21 and Utp1, since the co-expression of Utp21/Utp1 with either Utp12/Utp13 or Utp6/Utp18 results in a tetrameric complex (Fig. 2e,f). In addition, the heterodimeric Utp6/Utp18 does not directly interact with Utp12/Utp13 but only with the tetrameric Utp12/Utp13/Utp1/Utp21 complex, which suggests that the heterodimer Utp21/Utp1 is localized between the other two dimers (Supplementary Fig. 3c). Altogether these data suggest that UtpB can be subdivided into three heterodimers of which the Utp12/Utp13 subcomplex

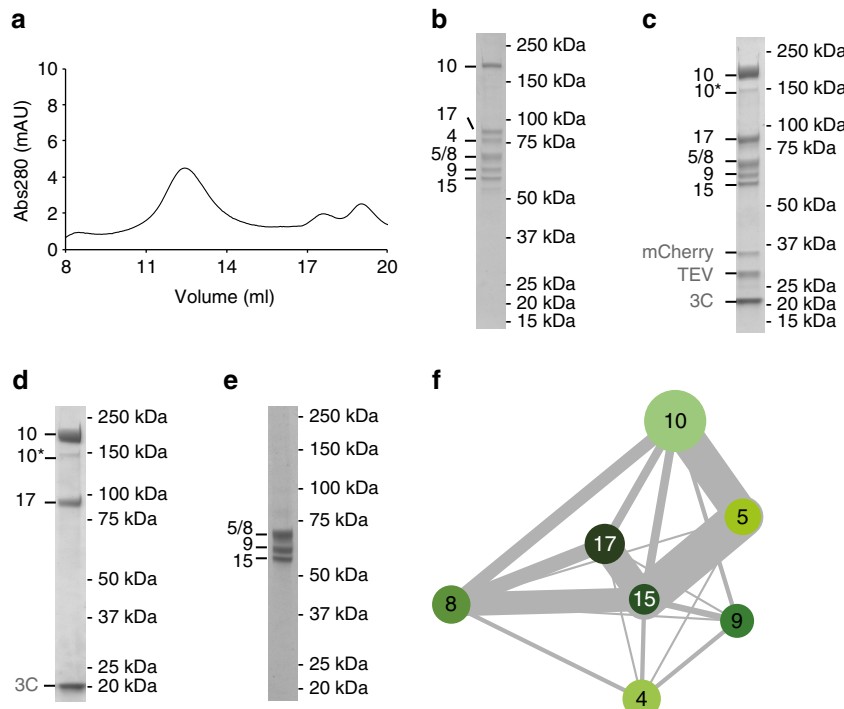

**Figure 1 | Purification and biochemical characterization of *Saccharomyces cerevisiae* UtpA.** (**a**) Size-exclusion chromatogram of endogenous UtpA and (**b**) corresponding visualization of the main peak fraction by 4–12% SDS-PAGE and Coomassie-blue staining. Co-eluting Utp proteins comprising UtpA (Utp4, Utp5, Utp8, Utp9, Utp10, Utp15, Utp17) are labelled by their respective number on the left. (**c,d**) SDS-PAGE analysis of UtpA subcomplexes resulting from purifications with buffers of different ionic strengths. Tagged UtpA (Utp10-3C-GFP and Utp15-TEV-mCherry) was purified at 200 mM NaCl on anti-GFP sepharose and incubated with buffers containing either 400 mM NaCl, yielding UtpAΔUtp4 (**c**), or 800 mM NaCl, yielding the Utp10-Utp17 dimer (**d**). 10* labels a degradation product of Utp10. TEV and 3C proteases (grey) were used for the elution and the removal of mCherry (grey). (**e**) SDS-PAGE analysis of the main peak fraction of co-eluting Utp5, Utp8, Utp9 and Utp15 on size-exclusion chromatography. Utp4, Utp5, Utp8, Utp9 and Utp15-TEV-mCherry were overexpressed in yeast and affinity purified. Utp4 dissociated from the complex during size-exclusion chromatography resulting in the elution of the heterotetramer shown. (**f**) Schematic representation of intersubunit DSS cross-links (grey lines) between UtpA subunits (circles coloured in different shades of green). The thickness of the lines connecting subunits reflects the number of cross-links shared between the subunits.

(Fig. 2c, Supplementary Fig. 3a,b) interacts via the central Utp21/Utp1 dimer (Fig. 2e) with Utp6/Utp18 (Fig. 2d,f, Supplementary Fig. 3c).

We investigated protein–protein interactions within the heterotetramer containing Utp12, Utp13, Utp21 and Utp1 by cross-linking and mass spectrometry. The results confirmed that this complex is composed of a pair of heterodimers with strong cross-links between Utp12 and Utp13, as well as Utp21 and Utp1 (Fig. 2g).

Combined with previous cross-linking and biochemical data of UtpB, this suggests that two heterodimers formed by Utp12 and Utp13, or Utp21 and Utp1 can interact to form a central core complex[14,16,17]. Utp6 and Utp18 were previously shown to form a separate element of the complex, with limited cross-links to the other four subunits[14,17,27].

Utp12, Utp13, Utp21 and Utp1 have a common architecture with an N-terminal tandem β-propeller followed by an α-helical region. Numerous intra-protein cross-links exist (Supplementary Fig. 2b), suggesting an intertwined architecture of tandem β-propellers of Utp1 and Utp13 similar to that previously observed for Utp21 (ref. 16).

**UtpB has an extended structure with separate modules**. To understand the structural basis of the cross-linking mass spectrometry analysis we performed negative-stain EM of UtpA and UtpB. While UtpA displayed a variety of conformations and was not suitable for 3D reconstruction (Supplementary Fig. 4),

endogenous UtpB purified from *Saccharomyces cerevisiae* was further analysed by random conical tilt 3D reconstruction (Fig. 3). To stabilize UtpB, gentle on-column glutaraldehyde cross-linking was performed (Supplementary Fig. 5), which increased the number of intact particles for subsequent studies as previously observed for other large multi-protein complexes[28,29].

The analysis of 2D class averages obtained by iterative stable alignment and clustering (ISAC)[30] revealed the presence of an elongated structural core, which is connected at one end to a second, smaller module (Fig. 3a and Supplementary Fig. 6a). The position of the smaller module relative to the core varied between different class averages, suggesting that it is tethered to the core through a flexible linker. To identify the position of Utp6 and Utp18, we used negative-stain EM to analyse a recombinantly expressed 400 kDa heterotetrameric complex consisting of Utp12, Utp13, Utp21 and Utp1. The resulting 2D class averages showed that Utp12, Utp13, Utp21 and Utp1 are sufficient for the formation of the structural core of UtpB, implying that the mobile second module corresponds to Utp6 and Utp18 (Fig. 3b and Supplementary Fig. 6b). We could independently show by size-exclusion chromatography that Utp18 is required for stable association of Utp6 with the core of UtpB (Supplementary Figs 3c and 7). These data suggest that a flexible linker corresponding to the N-terminal region of Utp18 (Supplementary Fig. 7d) connects the structural core of UtpB, which consists of Utp12, Utp13, Utp1 and Utp21, to a second module composed of the β-propeller of Utp18 and the HAT repeat of Utp6 (Fig. 3 and Supplementary Fig. 7). This model is also in agreement with previously published

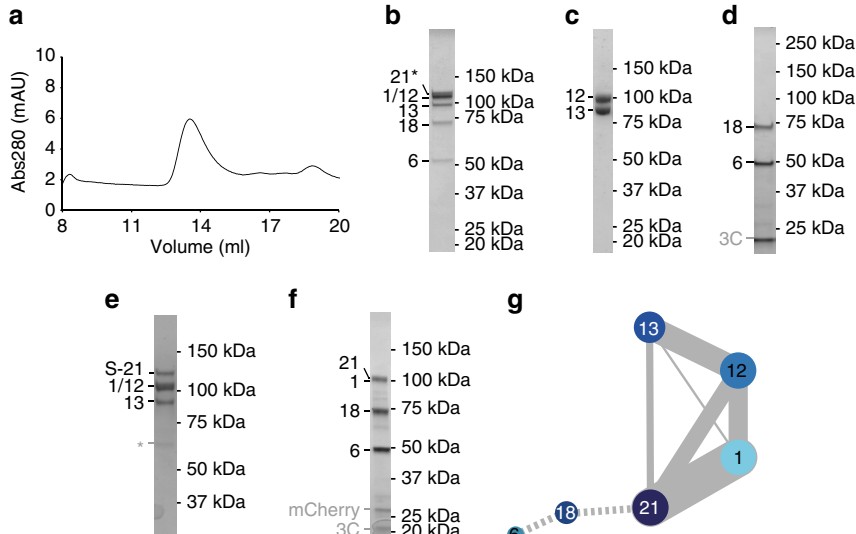

**Figure 2 | Purification and biochemical characterization of *Saccharomyces cerevisiae* UtpB.** (**a**) Size-exclusion chromatogram of endogenous UtpB and (**b**) corresponding visualization of the main peak fraction by 4–12% SDS-PAGE. Co-eluting Utp proteins comprising UtpB (Utp1, Utp6, Utp12, Utp13, Utp18 and Utp21) are labelled by their respective number on the left. * indicates that Utp21-3myc-TEV-mCherry-Flag was used as affinity purification tag and contains 3 C-terminal myc tags after TEV cleavage. (**c–f**) Purified recombinant subcomplexes of UtpB expressed in bacteria and analysed on a 4–12% SDS-PAGE gel. (**c**) Main peak fraction of a size-exclusion chromatography run with co-expressed and purified Utp12 and Utp13. (**d**) 3C protease elution of the Utp6/Utp18 pull-down control shown in Supplementary Fig. 3c. (**e**) Main peak fraction of a size-exclusion chromatography run of co-expressed and purified Utp12/Utp13/Utp21/Utp1. (S-21 indicates StrepII-Sumo-Utp21). * indicates a GroEL contamination. (**f**) 3C protease elution of co-expressed heterotetrameric Utp6/Utp18/Utp21/Utp1 purified by tandem affinity purification via His14-3C-Utp21 and mCherry-3C-Utp6. (**g**) Schematic representation of intersubunit DSS cross-links (grey lines) between UtpB subunits Utp12, Utp13, Utp21 and Utp1 (circles coloured in different shades of blue). The thickness of the lines connecting subunits reflects the number of cross-links shared between the subunits. Solid lines represent DSS cross-links determined in this study. Dashed lines indicate subunits Utp6 and Utp18 that were not included in the cross-link analysis.

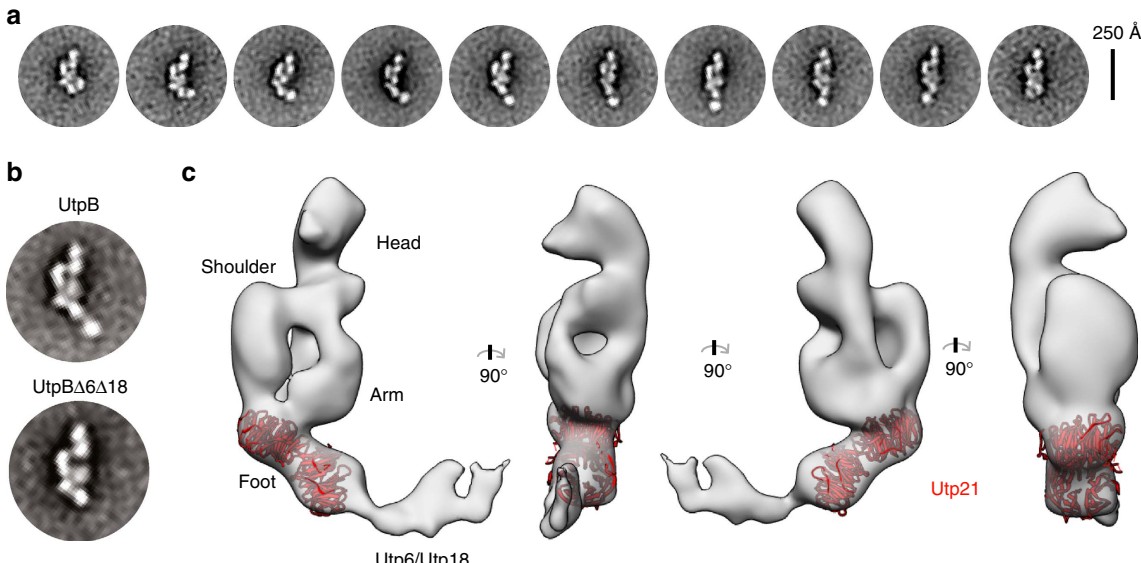

**Figure 3 | UtpB has an elongated structure with a mobile globular domain on one end.** (**a**) Representative ISAC[30] 2D class averages showing a tetrameric elongated body with a highly mobile domain that can sample a large range of conformations. (**b**) Comparative ISAC 2D class averages of endogenous UtpB and of recombinant UtpB lacking Utp6 and Utp18. (**c**) Different views of a random conical tilt reconstruction of UtpB (EMD-8223) show the elongated structure of the tetrameric core and weak density for the flexible domain at the tip of the foot. The manually docked double-β-propeller of Utp21 (pdb code 4nsx (ref. 16)) is shown as cartoon representation in red in the foot region of UtpB.

data showing a direct interaction between the N-terminal region of Utp18 (residues 100 to 190) with the N-terminal tandem β-propeller of Utp21 (ref. 16).

To obtain 3D structural information of the conformationally flexible UtpB, we acquired a large negative-stain EM tilt-pair data

set (Supplementary Fig. 8a,b). 2D classification into 50 classes resulted in 2D averages corresponding to several different conformations of UtpB, which represent different degrees of bending (Supplementary Fig. 8c). However, one predominant conformation can clearly be identified from these classes

(Supplementary Fig. 8d, groups 8, 13, 19, 33 & 47). Random conical tilt reconstructions of these classes resulted in similar volumes showing an elongated structure of about 230 Å in length and 90 Å in width (Supplementary Fig. 8d, blue volumes). In most reconstructions little or no density was observed for the mobile domain at one end of UtpB. To obtain the final volume, we combined the particles from the tilted specimens of these classes with 10% of the corresponding particles from the untilted specimen and performed angular refinement in SPIDER[31] (Fig. 3c and Supplementary Fig. 9). The final reconstruction has a resolution of 28 Å (Supplementary Fig. 9a). Weak, extended density can be observed for the mobile module, which is composed of Utp6/Utp18 (Fig. 3c and Supplementary Fig. 9c). Topologically the core structure of UtpB is defined by a head module, which is connected to the rest of the particle via a curved shoulder region. The lower part of the structure is composed of two lobes, which we refer to as arm and foot regions (Fig. 3c). The foot region further serves as connection to the flexible module, which is composed of Utp6/Utp18. Different conformations are observed for the flexible module, which can sample almost the entire region around the tip of the foot structure (Fig. 3a). Since previous biochemical experiments have shown that the tandem β-propeller domain of Utp21 can directly interact with an N-terminal segment including residues 100 to 190 of Utp18 (ref. 16), we have tentatively assigned the N-terminal tandem β-propeller domains of Utp21 to the foot region of the structure. This placement allows the N-terminal region of Utp21 to interact with the Utp6/Utp18 module, while the C-terminal region of Utp21 can interact with the other three subunits. On the basis of our previous biochemical characterization, the close interaction of Utp21 with Utp1 suggests that Utp1 contributes to the density observed in the lower part of the structure while Utp12 and Utp13 most likely contribute to the top of the structure.

This tentative assignment is consistent with the observation that Utp12 and Utp13 as well as Utp21 and Utp1 form heterodimers. It is further in very good agreement with protein cross-linking data that revealed that Utp21 is most readily cross-linked to Utp1, less to Utp12 and much less to Utp13 (Fig. 2g, Supplementary Figs 2b and 9c).

**UtpA and UtpB bind distinct sites of pre-rRNA and U3 snoRNA.** To determine the *in vivo* pre-rRNA-binding sites of both UtpA and UtpB, we applied CRAC. A tripartite tag consisting of a polyhistidine tag, a TEV cleavage site, and a protein A tag (HTP) was inserted at the genomic loci encoding subunits of the UtpA and UtpB complexes, under the control of the endogenous promoters. Following *in vivo* ultraviolet cross-linking in actively growing cells, covalently bound RNA-protein complexes were subjected to multistep purification. Complexes were initially purified by binding to IgG Sepharose followed by TEV protease elution. Bound RNAs were then partially digested followed by two denaturing purification steps; nickel affinity purification in the presence of 4 M guanidinium-HCl and SDS-PAGE. These steps specifically select RNA that was covalently cross-linked to the tagged protein subunit, which was identified by RT-PCR and Illumina sequencing. Hence only direct RNA-protein interactions are identified for a single protein subunit in a single experiment. By targeting multiple subunits of each complex, ensembles of CRAC profiles were obtained that show characteristic patterns for each complex (Fig. 4). Strikingly, all seven subunits of UtpA showed predominant cross-linking in the 5′ proximal region of the 5′ ETS (Fig. 4a), consistent with a key role in initiating the ribosome assembly process. The wild-type control showed only a prominent peak in the 25S rRNA, which has been seen in many experiments and represents a common contaminant[6,23,32].

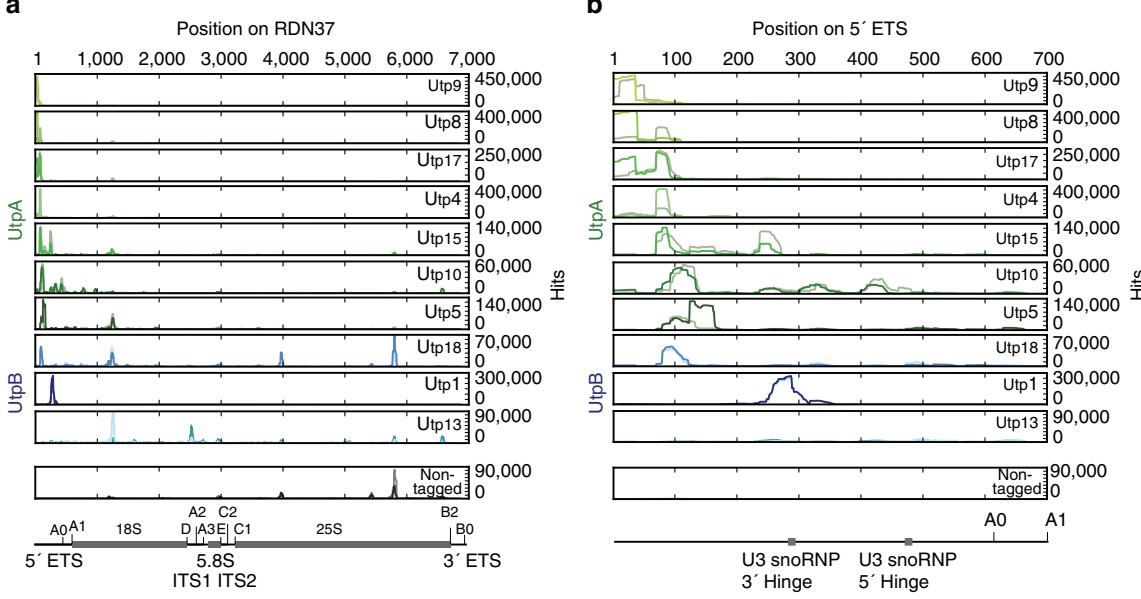

**Figure 4 | Binding sites of UtpA and UtpB within the 35S pre-rRNA.** (**a**) Sequences obtained from CRAC experiments with all UtpA subunits (in shades of green, duplicates in light green), selected UtpB subunits (in shades of blue, duplicates in light blue) and the non-tagged control (in grey, duplicate in light grey) were aligned to the rDNA locus (RDN37) and plotted as frequency of recovery (hits per million mapped reads) at each nucleotide position (indicated above all panels). Individual scales for the frequency of recovery are shown on the right of each subunit panel. The 35S pre-rRNA encoded by RDN37 is schematically depicted below the traces with cleavage sites (A0, A1, D, A2, A3, E, C2, C1, B2, B0), internal and external spacer regions (ITS1, ITS2, 5′ ETS, 3′ ETS) and ribosomal RNA (18S, 5.8S, 25S) indicated. (**b**) Expanded view of CRAC library hits on the 5′ ETS (nucleotides 1–700 of RDN37). Positions of U3 snoRNA base-pairing sites (3′ Hinge, 5′ Hinge) and pre-rRNA cleavage sites (A0, A1) are shown on a schematic representation of the 5′-ETS below.

Expansion of the 5′ ETS region (Fig. 4b) shows differences in the peak cross-linking sites for different components, suggesting the pathway of the pre-rRNA through the complex. Utp9 had the most 5′ proximal position with strong cross-linking only within the first 40 nt of the 5′ ETS. Utp8 and Utp17 also bind within the 5′ 40 nt but showed additional cross-linking around +90. Utp4 showed only the peak at +90, while Utp15 was cross-linked at this site and further 3′ around +250, close to the binding site for the U3 snoRNA 3′ hinge region. The large Utp10 protein showed peak cross-linking around +110, with weaker binding at sites from the 5′ end to around +500, strongly suggesting that it interacts with the pre-rRNA in an extended conformation. Finally, the peak of Utp5 cross-linking was seen around +130. Altogether these data reveal that the UtpA complex incorporates the 5′ end of the nascent pre-rRNA, with extensive interactions up to around +150.

For the UtpB complex, C-terminal tagging was unsuccessful for Utp12 and Utp6, whereas tagged Utp21 was not significantly cross-linked to RNA (data not shown). Reads were obtained for the remaining three UtpB subunits, Utp1, Utp13 and Utp18, which are either part of the core of UtpB (Utp1 and Utp13) or the mobile module (Utp18) (Figs 3 and 4). Utp18 bound around +90 in the centre of the core UtpA binding region. Utp1 was predominately cross-linked around the U3 snoRNA binding site at +280 whereas Utp13 showed peak cross-linking around cleavage site D at the 3′ end of the 18S rRNA. These binding sites indicate that the elongated and conformationally flexible UtpB complex brings together functionally important sites that are dispersed in the pre-rRNA sequence.

In addition, several proteins, notably Utp5, Utp15 and Utp18, showed peaks of cross-linking around +1,200 nt in 35S (+500 within the 18S rRNA). In the 18S rRNA secondary structure this

site lies close to the 'central pseudoknot', a key structural feature of the small ribosomal subunit, and may also be closely located in early pre-ribosomes.

The pre-rRNA cross-linking data place the UtpA and UtpB complexes in close proximity to binding sites for the U3 snoRNA and we therefore also analysed reads corresponding to this RNA (Fig. 5a). Notably, all UtpA and UtpB components showed U3 snoRNA cross-linking that was substantially (>10-fold) higher than the negative control. However, the cross-linking of Utp10 and Utp1 to U3 snoRNA was more than 10-fold higher than that of other UtpA or UtpB subunits. Since the read numbers in Fig. 5 are expressed as hits per million mapped reads, this reflects relatively strong cross-linking of these proteins to U3 snoRNA compared with the pre-rRNA (peak heights in Figs 4 and 5).

The U3 snoRNP has a pronounced domain structure, with a large, highly structured 3′ domain that binds the core snoRNA proteins including Nop56, Nop58, Nop1 (fibrillarin) and Rrp9[33]. The 5′ domain is relatively unstructured and contains pre-rRNA base-paring regions, including the 5′ and 3′ hinge regions and box A (Fig. 5b)[20–22]. Utp10 predominately cross-linked to the 3′ domain of U3 snoRNA, adjacent to major binding sites for the snoRNP proteins (Fig. 5b). In contrast, Utp1 cross-linked only over the 3′ hinge region in the 5′ domain of U3 snoRNA (Fig. 5b). This is consistent with the specific binding of Utp1 with the pre-rRNA target for U3 snoRNA base-pairing (Fig. 4b). A second UtpB subunit, Utp18, showed a low level of cross-linking to the 3′ hinge in U3 snoRNA (Fig. 5a and Supplementary Fig. 10).

Other Utp proteins all had low levels of reads within the large terminal stem of the U3 snoRNA 3′ domain; either on the 5′ side (Utp17) or 3′ side (Utp4, Utp5, Utp8, Utp13, Utp15, Utp18) (Supplementary Fig. 10b,c). Notably, no significant cross-linking was seen to the other experimentally

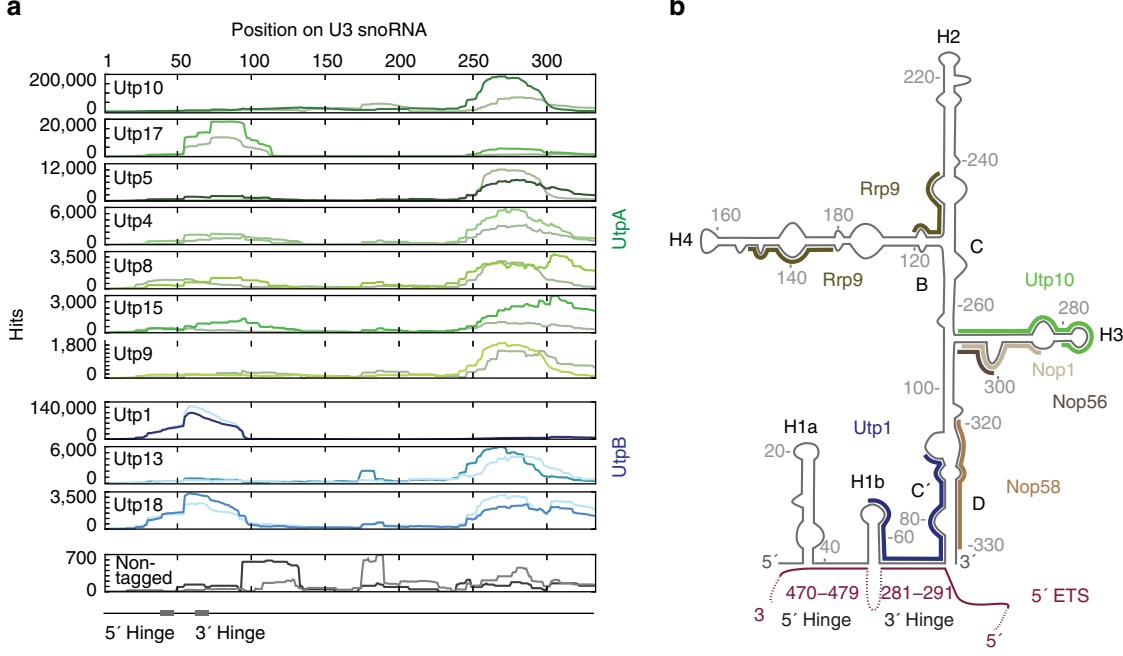

**Figure 5 | UtpA and UtpB contact the U3 snoRNA. (a)** CRAC library sequences of UtpA (in shades of green, duplicates in light green), UtpB (in shades of blue, duplicates in light blue) and the non-tagged control (in grey, duplicate in light grey) mapped to the spliced SNR17A (encoding the U3 snoRNA) are plotted as frequency of recovery (hits per million mapped reads) at each nucleotide position (indicated above all panels). Individual scales for the frequency of recovery are shown on the left of each subunit panel. Subunit panels are ordered by protein complex and further by their respective number of hits per million mapped reads. The positions of the 3′ and 5′ hinges that base-pair with the 5′ ETS are indicated below the traces. **(b)** Schematic secondary structure of U3 snoRNA (black) with base-paired 5′ ETS (purple) and CRAC-based binding sites for the U3 snoRNP proteins Rrp9, Nop1, Nop56 and Nop58 in different shades of brown, as determined previously[33], binding sites for Utp10 (green) from UtpA and Utp1 (blue) from UtpB. Helices are labelled with H and the conserved Box B/C/C′/D sequence elements are marked with their single letter respectively.

confirmed pre-rRNA-binding sites in the 5′ region of U3 snoRNA; the 5′ hinge and box A[20–22].

Altogether these data suggest a role for Utp10 within the UtpA complex in recruiting the U3 snoRNP, while Utp1 within the UtpB complex subsequently interacts with both RNA strands in the base-paired U3 snoRNA - pre-rRNA interaction at +280 in the 5′ ETS.

## Discussion

Here we provide first detailed insights into the structure and function of the large, multi-subunit protein complexes UtpA and UtpB. By combining biochemical and structural biology approaches with ensembles of RNA-protein cross-linking data, we determined the molecular organization of both complexes, their key RNA-protein contacts *in vivo* and the architecture of UtpB.

The absence of high-resolution structural information of many components of UtpA and UtpB has so far limited our understanding of these complexes. Here we show that the use of protein cross-linking and mass spectrometry in combination with biochemistry allowed the general architecture of UtpA and UtpB to be deciphered. While the bulk of inter-protein cross-links of UtpB are contained within the C-terminal domains of four UtpB subunits, a broader distribution of inter-protein cross-links is observed for subunits of UtpA (Supplementary Fig. 2).

RNA-protein interactions identified by CRAC correspond to a population of temporal states. However, previous EM analyses of 'Miller' chromatin spreads and analyses of metabolic labelling strongly indicate that early pre-ribosome assembly occurs co-transcriptionally[34,35]. Therefore, while we cannot distinguish sequential RNA-binding events of individual UtpA or UtpB subunits it seems likely that the 5′ to 3′ location of cross-linking sites will at least partially reflect the order of binding. In Fig. 6 we present a model, for the potential, sequential RNA binding of UtpA and UtpB subcomplexes. At the earliest stages of transcription, three subunits of UtpA (Utp8, Utp9 and Utp17) bind to nascent pre-rRNA at the very 5′ end while the remaining four subunits (Utp10, Utp4, Utp5 and Utp15) interact with nucleotides further downstream in the 5′ ETS (Fig. 6a). The elongated and flexible structure of UtpB suggests that this complex is ideally suited to bridge distinct RNA-binding sites. After an initial binding to the 5′ ETS via the flexible module containing Utp18 (Figs 3 and 6b), the core of UtpB interacts with both RNA strands of the duplex formed by the 5′ ETS and the 3′ hinge of U3 snoRNA via Utp1 (Fig. 6c). During later stages of

SSU processome assembly, the completion of the 18S rRNA and resulting structural changes enable Utp13 to interact with the 3′ boundary of the 18S rRNA (Fig. 6d,e). This model is in good agreement with our previous observation that a stage-specific assembly occurs during SSU processome formation[11], which was recently independently confirmed[36].

Our systematic analysis of RNA-protein cross-linking highlights that overlapping binding sites exist for different subunits of UtpA and UtpB both within the 5′ ETS as well as the U3 snoRNA (Figs 4 and 5). These may either reflect close spatial proximity within the SSU processome, or dynamic structural changes within the nascent SSU processome. Notably, the interactions of UtpA and UtpB with U3 snoRNA seem to be with specific sub-structures of this RNA, as no interactions were observed with helix 1a, the 5′ hinge or regions previously implicated in Rrp9 binding[33]. This observation supports a temporal order of U3 snoRNA binding with distinct sites in the pre-rRNA, in which the most 5′ interaction site, at +280 in the 5′ ETS, is bound by U3 snoRNA before the sites at +470 and within the 18S rRNA. The U3 snoRNA-5′ ETS interaction at +280 is required for subsequent pre-rRNA processing, but involves only a relatively short region of complementarity (11nt)[22]. We speculate that U3 snoRNP recruitment may be stimulated by UtpA via the Utp10-U3 snoRNA 3′ domain interactions, while specific U3 snoRNA-5′ ETS base-pairing may be facilitated by UtpB, via Utp1 bridging the interaction site.

Beyond the immediate early steps of eukaryotic ribosome assembly, the recent identification of Utp18 as an interaction partner of the RNA helicase and exosome cofactor Mtr4 suggests that Utp18 and hence UtpB could fulfil a dual role by both stabilizing the 5′ ETS and U3 snoRNA for productive ribosome assembly as well as recruiting Mtr4 for either degradation of 5′ ETS particles after A0/A1/A2 cleavage events or for degradation of defective ribosome assembly intermediates[37]. Intriguingly, early pre-ribosomal intermediates also contain Utp3/Sas10 and Lcp5—both of which carry putative exosome interaction domains based on their homology to Rrp47 (refs 11,38).

In contrast to prokaryotic ribosome assembly, eukaryotic ribosome assembly is characterized by a largely expanded network of RNA chaperones and quality control factors. Many of these factors—including UtpA and UtpB subunits—contain β-propellers and α-helical repeats as building blocks. This suggests that these factors come together to assemble a very large structural framework. This structure may give rise to the 'terminal balls' long observed on the 5′ termini of the nascent

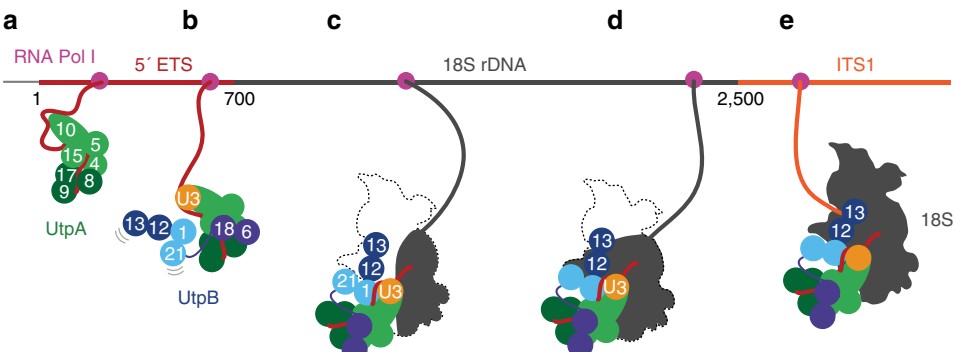

**Figure 6 | Schematic model of early co-transcriptional events in eukaryotic ribosome biogenesis.** (**a**) RNA polymerase I (pink) synthesizes the 5′ ETS (dark red), which is co-transcriptionally captured by several subunits of UtpA (shades of green). (**b**) More subunits of UtpA and Utp18 of UtpB (shades of blue) bind to the 5′ ETS. The U3 snoRNP (orange) is recruited to UtpA and UtpB. (**c**) The U3 snoRNP (orange) is base-paired at the 3′ hinge with the 5′ ETS and in close proximity to Utp1. (**d**) Transcription continues and more parts of the 18S rRNA (dark grey) is generated. (**e**) The completion of the SSU processome with a largely folded 18S enables Utp13 to bind in the vicinity of the 3′ end of the 18S rRNA.

pre-rRNA in 'Miller spreads' of rDNA chromatin. We speculate that this framework allows the large-scale organization—and reorganization—of the maturing pre-ribosomal particles[3].

## Methods

**Strains and media.** All yeast strains are derived from (BY4741, MATa; *his3Δ1*; *leu2Δ0*; *met15Δ0*; *ura3Δ0*). Standard techniques were used to integrate C-terminal affinity tags and integration of galactose-driven genes. Strains used are listed in Supplementary Data 1.

**Growth of cultures and cell lysis (UtpA and UtpB).** Yeast strains used for endogenous complex purifications, YSK43 or YSK47 (UtpB) and YSK32 (UtpA), were grown to saturation in YPD medium and collected by centrifugation at 4,000 *g*. Cell pellets were washed twice with ice-cold water and once with a volume of water supplemented with protease inhibitors (PMSF, Pepstatin A, E64) equal to the weight of each pellet. The final cell paste was flash-frozen as 'noodles' by pushing it through a syringe into liquid nitrogen. Cell disruption was performed by cryo-milling using a Retsch Planetary Ball Mill PM 100, and the cryo-ground powder was stored at −80 °C until further use.

To overexpress Utp4, Utp5, Utp8, Utp9 and Utp15-3myc-TEV-mCherry-3FLAG, a pre-culture of YSK122 was grown overnight in YP medium containing 2% (w/v) glucose at 30 °C. The pre-culture was used to inoculate a larger volume of YP medium supplemented with 2% (w/v) raffinose. Cells were grown at 30 °C in raffinose-containing medium until $OD_{600} = 0.9$. Expression of proteins under galactose promoters was induced by the addition of 2% (w/v) galactose overnight at 30 °C, and cells were collected and lysed as described above.

**Purification of endogenous UtpA for protein cross-linking.** 30 grams of cryo-milled powder from strain YSK32 were resuspended in 30 ml of binding buffer (50 mM HEPES-NaOH pH 7.6 (4 °C), 200 mM NaCl, 1 mM EDTA, 1 mM DTT, 10% glycerol) supplemented with protease inhibitors (PMSF, Pepstatin A, E64), DNAseI and RNaseA. The solution was incubated on ice for 30 min and cleared by centrifugation at 40,000 *g*, 4 °C for 30 min. The cleared lysate was incubated with 500 μl of anti-GFP nanobody-coupled sepharose for 4 h at 4 °C on a nutator and washed six times with binding buffer. Bound protein complexes were eluted by incubation with TEV and 3C protease for 150 min at 4 °C. Eluted protein complexes were further purified by size-exclusion chromatography (Superose 6 10/300 GL, GE Healthcare) in binding buffer lacking glycerol and DTT.

**Purification of subcomplexes of UtpA.** *UtpAΔUtp4, Utp10-Utp17.* 11 grams of cryo-milled YSK32 powder were resuspended in 44 ml binding buffer (50 mM HEPES-NaOH pH 7.6 (4 °C), 200 mM NaCl, 1 mM EDTA, 1 mM DTT, 10% glycerol) supplemented with protease inhibitors (PMSF, Pepstatin A, E64). The lysate was cleared for 30 min at 40,000 *g*, 4 °C. 560 μl of anti-GFP nanobody-coupled sepharose was added to the supernatant and incubated for 3.5 h at 4 °C. Beads were washed twice with binding buffer and distributed in 14 tubes (40 μl of beads each). The aliquots were washed and incubated with binding buffer lacking glycerol and containing concentrations of NaCl ranging from 200 mM to 800 mM in 100 mM steps (50 mM HEPES-NaOH pH 7.6 (4 °C), 200–800 mM NaCl, 1 mM EDTA, 1 mM DTT) for 7 h at 4 °C. Protein complexes were eluted by 3C protease cleavage overnight at 4 °C. Elutions were spun at 15,000 *g* for 6 min and supernatants were analysed on a 4–12% SDS-PAGE gel by Coomassie-blue staining.

*UtpAΔUtp4ΔUtp10ΔUtp17.* 20 grams of cryo-milled YSK122 powder were resuspended with 80 ml of binding buffer (50 mM HEPES-NaOH pH 7.6 (4 °C), 100 mM NaCl, 1 mM EDTA, 1 mM DTT, 10% glycerol) supplemented with protease inhibitors (PMSF, Pepstatin A, E64), DNAseI and RNaseA. The lysate was cleared by centrifugation at 40,000 *g*, 4 °C for 30 min. 800 μl of anti-mCherry nanobody-coupled sepharose were added to the supernatant and incubated for 4 h at 4 °C. Beads were washed six times with binding buffer. The protein complex was eluted by TEV cleavage at 4 °C for 150 min. Endogenous UtpA was removed through Utp10-sbp-H14-3C-GFP by incubating the elution with anti-GFP nanobody-coupled sepharose for 30 min. The flow through was stored overnight at 4 °C and centrifuged for 10 min at 15,000 rpm at 4 °C before injection on a Superose 6 Increase 10/300 GL column (GE Healthcare) equilibrated with 50 mM HEPES-NaOH pH 7.6 (4 °C), 200 mM NaCl, 1 mM EDTA.

**Purification of UtpB for random conical tilt reconstruction.** 21 grams of cryo-milled powder from strain YSK43 were resuspended with 42 ml of buffer K (100 mM CHES-NaOH pH 9 (4 °C), 300 mM potassium acetate, 5 mM EDTA, 1 mM DTT) supplemented with protease inhibitors (PMSF, Pepstatin A, E64), DNAseI and RNaseA. The solution was incubated on ice for 10 min and then cleared by centrifugation at 40,000 *g*, 4 °C for 30 min. The cleared lysate was incubated with 666 μl anti-mCherry nanobody-coupled sepharose for 4 h at 4 °C on a nutator and washed four times with buffer K. Protease cleavage was performed in 600 μl of 50 mM HEPES-NaOH pH 7.8 (4 °C), 200 mM NaCl, 1 mM EDTA and

2 μM TEV protease for 2 h on a nutator. The supernatant was collected, centrifuged for 15 min at 15,000 rpm at 4 °C before injection on a Superose 6 Increase 10/300 GL (GE Healthcare).

**On-column glutaraldehyde cross-linking.** The complex was on-column glutaraldehyde cross-linked[28] in 50 mM HEPES-NaOH pH 7.7 (4 °C), 200 mM NaCl, 1 mM EDTA as follows: 500 μl of a 0.25% glutaraldehyde solution was pre-injected onto a size-exclusion column (Superose 6 Increase 10/300 GL, GE Healthcare) and run for 20 min at a flow rate of 0.25 ml/min. The run was paused, the injection loop vigorously washed, UtpB injected, and the run was continued at the same flow rate. The on-column cross-linking procedure was optimized with UtpB purified using affinity tags on different subunits (YSK43 and YSK47, Supplementary Data 1). UtpB from both strains exposed to the pre-injected glutaraldehyde bolus exhibited the same elution behaviour as the non cross-linked complex (Supplementary Fig. 5a) and the protein complex eluted at 13.14 ml.

**Purification of UtpB and subcomplexes from bacteria.** Standard techniques were used to clone genes of UtpB subunits from *Saccharomyces cerevisia* BY4741 genomic DNA into pRSFDuet1- or pETDuet1-derivatized expression plasmids. Constructs used for bacterial expression of UtpB are listed in Supplementary Data 2. The plasmid-containing His14-3C-Utp12/Utp13/StrepII-Sumo-Utp21/Utp1 was derived by combining plasmid pSKA048 (backbone, cut with AgeI) with a PCR amplified insert from pSKA052 via compatible NgoMIV/AgeI sites (The PCR introduced NgoMIV sites in front of T7 and after T7 terminator). This resulted in a construct containing His14-3C-Utp12/Utp13/StrepII-Sumo-Utp21/Utp1.

All constructs were expressed in *E. coli* RIL plasmid-containing cells. For co-expression of pSKA041-His-3C-Utp21/Utp1 with pSKA056-mCherry-3C-Utp6/Utp18 competent *E. coli* RIL cells were prepared with a pre-transformed plasmid pSKA041. Protein expression was induced by addition of 1 mM IPTG at an $OD_{600}$ between 0.6 and 1.0. Cells were grown overnight at 20 °C, collected and flash-frozen as cell pellet or used immediately. Cell pellets were resuspended in Lysis and Wash buffer consisting of 50 mM HEPES-NaOH pH 7.6 (4 °C), 300 mM NaCl supplemented with protease inhibitors (PMFS, Pepstatin A, E64), DNAse and Sumo protease in the case of pSKA061 (no second affinity step was required). Cell lysis was performed using a cell disruptor (TS5, Constant Systems), and the cell lysate was cleared by centrifugation at 40,000 *g*, 4 °C for 60 min. The cleared lysate was passed through a 5-ml HisTrap column (GE Healthcare) and washed by increasing the imidazole concentration to 70 mM. Proteins were eluted in 250 mM imidazole and fractions containing protein were pooled, supplemented with 3C protease and dialyzed overnight against 50 mM HEPES-NaOH pH 7.6 (4 °C), 300 mM NaCl, 0.5 mM EDTA, 0.5 mM DTT. To remove uncleaved protein and the cleaved affinity tag, the solution was passed once more over the 5-ml HisTrap column. The flow through was concentrated and further purified on a HiLoad 16/600 Superdex 200 prep grade (GE Healthcare) in 50 mM HEPES-NaOH pH 7.6 (4 °C), 300 mM NaCl, 1 mM EDTA, 1 mM DTT. Proteins were supplemented with 10% glycerol, flash-frozen and stored at −80 °C.

**Analytical size-exclusion chromatography.** Size-exclusion chromatography analysis of bacterial subcomplexes of UtpB (Supplementary Figs 3a and 7a) was performed in 50 mM HEPES-NaOH pH 7.7 (4 °C), 300 mM NaCl, 1 mM EDTA using a Superose 6 Increase 10/300 GL column (GE Healthcare) at a flow rate of 0.5 ml/min. 500 μl samples of 2 μM UtpBΔUtp6ΔUtp18 were injected either alone, with 4 μM Utp6/Utp18 or with 4 μM Utp6. For the Utp12/Utp13 size-exclusion chromatography run, 500 μl of 0.25 μM Utp12/Utp13 were injected. Peak fractions were analysed on 4–12% SDS-PAGE and Coomassie-blue staining (Supplementary Figs 3b and 7b,c).

**UtpB pull-down experiments.** The pull-down assay (Supplementary Fig. 3c) was performed using Nickel Sepharose 6 fast flow beads (GE Healthcare). His-tagged Utp6 was used as bait either alone or co-expressed with Utp18. Proteins were pre-incubated at a concentration of 1 μM in 100 μl for 10 min on ice in buffer PD (50 mM Hepes-NaOH, pH 7.7 (4 °C), 300 mM NaCl, 70 mM Imidazole) and applied on 20 μl of washed and dried beads. After 20 min incubation on ice, beads were washed 5 times with buffer PD and carefully dried. Proteins were eluted with 45 μl buffer PD supplemented with 2 μM 3C protease. The supernatant was removed and analysed by 4–12% SDS-PAGE and Coomassie-blue staining.

**DSS cross-linking of UtpA.** Peak fractions of size-exclusion chromatography-purified UtpA (in 50 mM HEPES-NaOH pH 7.6, 200 mM NaCl, 1 mM EDTA) were pooled (total volume 2 ml) and split into 200 μl cross-linking reactions. To each 200-μl aliquot 0.8 μl of DiSuccinimidylSuberate (DSS; 50 mM, 1:1 molar ratio mixture of DSS-H12 and DSS-D12, Creative Molecules Inc.) was added to yield a final DSS concentration of 0.2 mM. Samples were incubated at 25 °C for 30 min with 400 r.p.m. constant shaking. The cross-linking reaction was quenched with 50 mM ammonium bicarbonate. Cross-linked samples were precipitated by adding methanol to a final concentration of 90% and overnight incubation at −80 °C.

Precipitated cross-linked UtpA was collected in one tube by repeated centrifugation of the precipitated solution at 21,000 g, 4 °C for 15 min. The resulting pellet was washed once with 1 ml cold 90% methanol, air-dried and resuspended in 50 μl of 1X NuPAGE LDS buffer (Thermo Fisher Scientific, #NP0007).

**DSS cross-linking of UtpBΔUtp6ΔUtp18.** The cross-linking reaction of UtpBΔUtp6ΔUtp18 (size-exclusion purified in buffer 50 mM HEPES-NaOH pH 7.7 (4 °C), 300 mM NaCl, 1 mM EDTA) was performed in a total reaction volume of 150 μl at room temperature using different DSS concentrations (0.2, 0.4 and 0.8 mM 1:1 molar ratio mixture of DSS-H12 and DSS-D12, Creative Molecules Inc.) at a protein concentration of 0.2 mg/ml. The reaction was stopped after 30 min with 50 mM ammonium bicarbonate and supplemented with NuPAGE LDS (1X final). 10 μl of the reaction was analysed by 4–12% SDS-PAGE.

**Mass spectrometry analysis of UtpA and UtpBΔUtp6ΔUtp18.** DSS cross-linked UtpA or UtpB complexes in LDS buffer were reduced with 25 mM DTT, alkylated with 100 mM 2-chloroacetamide, separated by SDS-PAGE using several lanes of a 4–12% Bis-Tris gel, and stained with Coomassie-blue. The gel region corresponding to the cross-linked complexes was sliced and digested in-gel overnight with trypsin to generate cross-linked peptides. After digestion, the peptide mixture was acidified and extracted from the gel as previously described[39,40]. Analyses by LC-ESI-MS were performed either directly on the extracted peptides or following fractionation by size-exclusion chromatography[41] or high pH reverse-phase chromatography. Peptides were loaded onto an EASY-Spray column (Thermo Fisher Scientific, either ES800: 15 cm × 75 μm ID, PepMap C18, 3 μm or ES801: 15 cm × 50 μm ID, PepMap RSLC C18, 2 μm) via an EASY-nLC 1000. MS and MS/MS analyses were carried out on a Q Exactive Plus mass spectrometer (Thermo Fisher Scientific). MS/MS analyses of the top 8 precursors in each full scan used the following parameters: resolution: 17,500 (at 200 Th); AGC target: $2 \times 10^5$; maximum injection time: 800 ms; isolation width: 1.4 m/z; normalized collision energy: 29%; charge: 3–7; intensity threshold: $2.5 \times 10^3$; peptide match: off; dynamic exclusion tolerance: 1,500 mmu. Cross-linked peptides were identified from mass spectra by pLink[17]. All spectra reported here were manually verified[39,40].

All cross-links found for UtpA and UtpB are shown in Supplementary Data 3 and Supplementary Data 4. Figures were prepared using xiNET[42] (Figs 1f and 2g, and Supplementary Fig. 2).

**Ultraviolet cross-linking and high-throughput analysis of cDNA (CRAC).** Yeast strains that were actively growing in SD –TRP medium at 30 °C with an $OD_{600}$ of 0.5, were irradiated at 254 nm UV for 100–110 s as described[24]. Purification of RNA-protein complexes and RT-PCR amplification of associated RNA fragments was performed as described[33]. cDNA libraries were sequenced on an Illumina HiSeq2500 at Edinburgh Genomics, University of Edinburgh. Illumina sequencing data were aligned to the yeast genome using Novoalign (http://www.novocraft.com). Bioinformatics analyses were performed as described using PyCRAC[43,44].

**Negative-stain EM analysis of UtpA.** Purified UtpA was applied to glow-discharged home-made carbon-coated copper grids and negatively stained with 0.75% uranyl formate as previously described[45]. Images were recorded on a Philips CM10 operated at an acceleration voltage of 100 kV equipped with a XR16-ActiveVu (AMT) camera at a nominal magnification of 52,000 × and a calibrated pixel size of 2.8 Å at the specimen level.

**Negative-stain EM analysis of UtpB lacking Utp6 and Utp18.** Reconstituted UtpB lacking Utp6 and Utp18 was on-column cross-linked as endogenous UtpB, applied to glow-discharged carbon-coated 200 mesh copper grids (Electron Microscopy Sciences, CF200-Cu) and stained with 2% uranyl acetate. 98 micrographs were collected at a nominal magnification of 49,000 at a defocus of −1.6 μm on a Tecnai G2 spirit operated at 120 kV and a resulting pixel size of 2.17 Å at the specimen level. 8,171 particles were manually selected using e2boxer.py and windowed into 230 × 230-pixel images. The particles were subjected to ISAC, specifying 50 images per group and a pixel error threshold of 0.7. 23 generations yielded 171 classes, accounting for 3,343 particles (41% of the entire data set; Supplementary Fig. 6b).

**Negative-stain EM and 3D reconstruction of UtpB.** Endogenous UtpB was applied to glow-discharged home-made carbon-coated copper grids and negatively stained with 0.75% uranyl formate as previously described[45]. Images were recorded on a Tecnai T12 operated at an acceleration voltage of 120 kV. Tilt pairs were recorded with a Gatan 4k CCD camera under low-dose conditions at a calibrated magnification of 57,555 × and a resulting pixel size of 2.61 Å at the specimen level. A total of 420 tilt-pair micrographs at a defocus of −1.6 μm were collected (micrograph pairs 1–159 at tilt angles of 50° and 0°, the remaining pairs at tilt angles of 60° and 0°).

26,510 particle pairs were manually selected using e2RCTboxer.py[46] and tilt angles were determined using tiltpicker[47]. Particles were windowed into 192 × 192-pixel images and normalized. The particle images from the untilted specimens were subjected to ISAC, specifying 50 images per group and a pixel error threshold of 0.7. 14 generations yielded 598 classes, accounting for 9,077 particles (34% of the entire data set; Supplementary Fig. 6a). In parallel, the particle images were rotationally and translationally aligned, and subjected to 10 cycles of multi-reference alignment using SPIDER[31]. Each round of multi-reference alignment was followed by K-means classification specifying 50 output classes (Supplementary Fig. 8c). The references used for the first multi-reference alignment were randomly chosen from the particle images. For selected classes, 3D reconstructions were then calculated with the particle images from the tilted specimen using the back-projection and back-projection refinement procedures in SPIDER (Supplementary Fig. 8d). The final volume of UtpB (Fig. 3c, Supplementary Fig. 9b) was obtained by combining 5 classes (groups 8, 13, 19, 33 and 47, Supplementary Fig. 8d: blue volumes), and by using the angular refinement procedure in SPIDER. This density map included 3,517 particles selected from images of the tilted specimen and 319 particles from images of the untilted specimen. All volumes were low-pass filtered to 20 Å, and the noise surrounding the volume was removed using the auto mask procedure in e2proc3d.py (ref. 46).

**Data availability.** The negative-stain EM map of the Saccharomyces cerevisiae UtpB complex has been deposited in the Electron Microscopy Data Bank under the accession code EMD-8223. All sequence data have been deposited with the Gene Expression Omnibus (GEO) database (http://www.ncbi.nlm.nih.gov/geo/) under the accession number GSE79950. The data that support the findings of this study are available from the corresponding author on request.

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

## Acknowledgements

We thank Devrim Acehan (Electron Microscopy Center at The Rockefeller University, New York), Zongli Li (Harvard Medical School, Boston) for technical assistance with electron microscopy and Brian D. Dill (Proteomics Resource Center, The Rockefeller University, New York) for protein identification by mass spectrometry. M.H. was supported by a European Molecular Biology Organization (EMBO) Short-term fellowship (ASTF 352-2015) and the Rockefeller University Abroad Program. J.B. is supported by an EMBO long-term fellowship (ALTF 51-2014). K.H.K. is supported by a postdoctoral fellowship from the Canadian Institutes of Health Research. B.T.C. is supported by grants PHS GM103314 and PHS GM109824. E.P. and David T. were supported by the Wellcome Trust (097383), C.D.-F. was supported by an FEBS long-term fellowship, H.D.-D. was supported by the BBSRC (BB/F010656/1). M.C.-M. is supported by a scholarship from Fonds de Recherche du Québec–Santé (FRQ-S). S.K. is supported by the Robertson Foundation, the Alfred P. Sloan Foundation, the Irma T. Hirschl Trust, the Alexandrine and Alexander L. Sinsheimer Fund and the Human Frontier Science Program.

## Author contributions

M.H. purified and characterized UtpA. M.H. and E.P. performed ultraviolet cross-linking experiments of UtpA subunits. E.P. performed ultraviolet cross-linking experiments of UtpB subunits. M.H., H.D.-D. and C.D.-F. Analysed the ultraviolet cross-linking data. J.B. purified and characterized UtpB, and with help from Do.T., H.K. and T.W. performed electron microscopy and random conical tilt reconstruction. K.M., Y.S. and B.C. performed mass spectrometry data collection and analysis of cross-linked UtpA and UtpB. S.K. conceived the study, and together with M.H., J.B. and Da.T. wrote the manuscript. M.C.-M. provided reagents and assisted with data analysis. All authors analysed the data and edited the manuscript.

## Additional information

**Competing financial interests:** The authors declare no competing financial interests.



