## [Peer Review File · Nature Communications]

Response to reviewers

Reviewer #1 (Remarks to the Author):

“I have carefully read all reviewers' comments and believe that the authors have addressed most of them adequately.”

We thank reviewer 1 for this positive assessment of our work.

“I would suggest that they include more info in response to comment three of reviewer #3 about stoichiometry. I think that super-stoichiometric recovery of the tagged protein most likely reflects that what is recovered is a mixture of complexes containing that protein, plus some fraction that is the monomeric tagged protein not bound to other proteins.”

We agree with reviewer 1 that for example in Figure 2d and 2f, super-stoichiometric Utp6 is present since these samples also contain free tagged protein. We have added a sentence in the manuscript (page 6), which states:

“Super-stoichiometric amounts of tagged Utp6 were observed in some experiments (Fig. 2d, f), indicating the recovery of monomeric tagged protein in addition to the stoichiometric complex.”

Reviewer #3 (Remarks to the Author):

“In this revision, the authors have addressed my comments and suggestions. An ideal way to assess the quality of cross-linking data is to map part of the results to what is known, the crystal structure of Utp21. However, the lack of high resolution structural information makes it infeasible, which underlines the power of cross-linking approach. If the authors have additional space, I suggest that they add a couple of sentences to highlight the novel discovery enabled by cross-linking approach in their discussion.”

We thank reviewer 3 for this suggestion and have added two sentences in the discussion, which highlight the key principles of what we have observed based on our cross-linking and mass spectrometry data:

“The absence of high-resolution structural information of many components of UtpA and UtpB has so far limited our understanding of these complexes. Here we show that the use of protein cross-linking and mass spectrometry in combination with biochemistry allowed the general architecture of UtpA and UtpB to be deciphered. While the bulk of inter-protein cross-links of UtpB are contained within the C-terminal domains of four UtpB subunits, a broader distribution of inter-protein cross-links is observed for subunits of UtpA (Supplementary Fig. 2).”

“I support its publication on Nature Communications.”

We thank reviewer 3 for this positive assessment of our work.

Reviewer #4 (Remarks to the Author):

“While the authors have addressed many of the minor reviewer comments in the revised manuscript, it was disappointing to see that the primary concerns from the original manuscript were not adequately addressed. The inclusion of a micrograph of UtpA and a statement that it is not suitable for 3D reconstruction does little to improve the manuscript.”

We have included a micrograph of UtpA to answer why we did not present a structure of this complex. The micrograph further shows that several different conformations can be observed for this complex.

“And there remains a question of stoichiometry and localization of individual components of UtpB.”

As pointed out in our response to reviewer 1, we have added a sentence in the manuscript, which explains why super-stoichiometric amounts of Utp6 are observed in two cases.

“Particularly, the inclusion of localized proteins within the UtpB complex would help to tie all of the results together into a more comprehensive study. As it is, the point made by reviewer 2 that the multiple results are somewhat disjointed and not integrated remains a valid concern.”

In this manuscript we focus on the interactions of UtpA and UtpB with pre-ribosomal RNA *in vivo*. These data have been integrated with results obtained by biochemistry as well as cross-linking and mass spectrometry to elucidate how UtpA and UtpB can act upon their pre-ribosomal RNA substrate. Here we have used a low-resolution envelope of UtpB (at 28 Angstroms) to indicate that a flexible linker separates two functional modules within UtpB.

We agree with reviewer 4 that ultimately an atomic structure (by cryo-EM or X-ray crystallography) will provide a comprehensive picture of the intricate connectivity of all UtpB subunits.

REVIEWERS' COMMENTS:

Reviewer #1 (Remarks to the Author):

I have carefully read all reviewers' comments and believe that the authors have addressed most of them adequately. I would suggest that they include more info in response to comment three of reviewer #3 about stoichiometry. I think that super-stoichiometric recovery of the tagged protein most likely reflects that what is recovered is a mixture of complexes containing that protein, plus some fraction that is the monomeric tagged protein not bound to other proteins.

Reviewer #3 (Remarks to the Author):

In this revision, the authors have addressed my comments and suggestions. An ideal way to assess the quality of cross-linking data is to map part of the results to what is known, the crystal structure of Utp21. However, the lack of high resolution structural information makes it infeasible, which underlines the power of cross-linking approach. If the authors have additional space, I suggest that they add a couple of sentences to highlight the novel discovery enabled by cross-linking approach in their discussion.

I support its publication on Nature Communications.

Reviewer #4 (Remarks to the Author):

While the authors have addressed many of the minor reviewer comments in the revised manuscript, it was disappointing to see that the primary concerns from the original manuscript were not adequately addressed. The inclusion of a micrograph of UtpA and a statement that it is not suitable for 3D reconstruction does little to improve the manuscript. And there remains a question of stoichiometry and localization of individual components of UtpB. Particularly, the inclusion of localized proteins within the UtpB complex would help to tie all of the results together into a more comprehensive study. As it is, the point made by reviewer 2 that the multiple results are somewhat disjointed and not integrated remains a valid concern.

Response to Reviewers

Reviewer #1 (Remarks to the Author):

Previously, it has been shown by the authors and others that early stages of ribosome assembly in yeast involve hierarchical assembly of protein complexes UtpA, UtpB, Mpp10 and the U3 snoRNP with precursor RNA destined for the small ribosomal subunit. This is a very nicely written manuscript describing analysis of the protein topology of the UtpA and B protein complexes and their interactions with rRNA as well as with the U3 snoRNA. The authors purify the native A and B complexes using affinity tags and gel exclusion chromatography. The A complex contains seven proteins while the B complex has six. Pairwise interactions between these proteins were discerned by salt washes, coexpressing protein pairs in yeast, and protein-protein cross linking. A low resolution structure of the B complex was obtained by EM, revealing an elongated structure with Utp12 and 13 at one end, Utp1 and 21 in the middle, and Utp18 and 6 at the other end, in a flexible module. Binding sites to pre-rRNA and U3 snoRNA were mapped by UV cross linking.

Together, these results led the authors to propose a working model for early stages of assembly that should be extremely useful for those studying ribosome biogenesis. UtpA binds to sequences at the 5' end of the 5'ETS followed by UtpB functioning as flexible, elongated bridge to recruit the U3 snoRNP and bring together the ends of pre-25S pre-rRNA.

The authors have done an excellent job to describe their work.

We thank reviewer 1 for this positive assessment of our work.

Did protein protein cross linking experiments reveal interactions between the UtpA and UtpB complex proteins?

In this study we have only investigated protein-protein interactions within UtpA and within UtpB. No experiments were performed to study interactions between the two complexes in the contexts of the 5'ETS particle or the SSU processome as described in our previous work (Chaker-Margot et al. 2015). While such studies are ongoing, these will require more time and will be published in future.

Figure 2: What are the fainter bands seen in figure 2d,e,and f?

We have updated this figure. A separate pull-down experiment of Utp6 and Utp18 was repeated for panel d, which again shows that these two proteins interact. This was part of a larger experiment displayed in the new Supplementary Figure 3, which addresses some of the interactions within UtpB. We have further added labels for a minor GroEL contaminant (panel e) and for mCherry and 3C protease (panel f).

Figure 5: Is the 5' ETS really beige?

We have changed the figure legend to indicate that the 5'ETS is displayed in purple.

Reviewer #2 (Remarks to the Author):

The authors of this manuscript describe interaction studies and structural analysis of two subcomplexes, named UtpA and UtpB, which are involved in the assembly of ribosomes in yeast. The purification of the complexes is described, followed by identification of interactions between protein components by crosslinking analysis. The authors present a partial cryoEM structure of one of the complexes, UtpB and map the available crystal structure of one of the proteins into the densities. In addition, UV crosslinking is performed with several of the protein components of the UtpA and UtpB complexes and their binding sites on ribosomal RNAs and the snoRNA U3 are mapped. The manuscript reports a large body of work that is presented well in the context of the literature.

We thank reviewer 2 for this positive assessment of our work.

The main novelty of the work lies in the density map presented for the UtpB complex. However, the composition of the complexes as well as many of the interactions have been reported previously and the manuscript falls short of reporting the more detailed molecular interfaces and lacks functional aspects of the reported interactions. Unfortunately, hardly any structural information about the proteins that are part of the complex is available; only the structure of part of Utp21 could be fitted in the densities. In addition, the UV crosslinking data do not reveal novel information on the mechanistic of early ribosome assembly and this dataset is somewhat disjoint from the structural data and the information is not integrated beyond a very schematic model. Without the addition of substantial datasets on the function or regulation of these interactions, the manuscript seems more suitable for a specialised journal.

The main novelty of this work is the synergistic use of ensembles of RNA-protein cross-linking data with negative-stain electron microscopy. Together our data reveal that the UtpA complex binds to the 5' end of pre-ribosomal RNA and that the modular and flexible UtpB complex (of which we present the first structure) has distinct binding sites in functionally important regions of the nascent pre-ribosomal particle.

Reviewer #3 (Remarks to the Author):

In this paper, the authors studied the molecular architecture of UtpA and UtpB complexes in ribosome small subunit processome, and their binding sites on pre-ribosomal RNA. The complexes and subcomplexes were characterized by multiple biochemical approaches, including gel-filtration chromatography, chemical cross-linking and mass spectrometric analysis, negative stain EM, UV cross-linking and sequencing. Consistent results on subunit interface mapping were generated from various approaches. Overall, the experiments were carefully designed and executed to produce biologically important results. The manuscript is suitable for publication on Nature Communications.

We thank reviewer 3 for this positive assessment of our work.

Here are a few minor points that the authors may consider in revision:

1. It seems that only Utp2, Utp1, Utp12 and Utp13 were included in chemical cross-linking study (page 6, 2nd paragraph). Is there a reason Utp18 and UtpG6 were not included? The UtpB complex seems stable enough for EM studies.

The focus of our cross-linking and mass spectrometry analyses was on the more rigid body of UtpB. Hence the heterotetramer formed by Utp21, Utp1, Utp12 and Utp13 was used for our cross-linking and mass spectrometry experiment.

2. In EM density assignment (page 8), Utp21 was placed to the foot region, based on "biochemical and mass spectrometry experiments have shown that the tandem beta-propeller domain of Utp21 can interact with an N-terminal segment of Utp18". It is not clear which biochemical and MS experiments led to this conclusion.

We thank reviewer 3 for pointing out the lack of clarity in this sentence. We have changed this sentence to precisely indicate which experimental data were used:

"Since previous biochemical experiments have shown that the tandem β -propeller domain of Utp21 can directly interact with an N-terminal segment including residues 100 to 190 of Utp18¹⁶, we have tentatively assigned the N-terminal tandem β -propeller domains of Utp21 to the foot region of the structure."

We placed Utp21 in its current position in the EM density envelope based on four lines of evidence:

1. From comparison of the 2D averages of intact UtpB and UtpB lacking Utp6 and Utp18 (Figure 3b), it is clear that Utp18/Utp6 corresponds to the flexible module that extends from the foot structure of UtpB. Since the Utp18/Utp6 density does not connect to any other part of the complex, proteins that directly interact with Utp18/Utp6 have to be located in the foot structure.
2. Previously published data by Keqiong Ye's group showed that the region of residues 100-190 of Utp18, which is positioned just upstream of its WD40 repeat, is sufficient to directly interact with Utp21 (Zhang et al. PLoS One 2014, Figure 1C).

3. Since the flexible module formed by Utp18/Utp6 samples the space around the foot region (Figure 3A), the Utp18/Utp6 dimer appears to interact directly with this part of the structure.
4. The data in new Supplementary Figure 3 (panel c) show that the Utp12/Utp13 dimer requires the presence of the Utp21/Utp1 dimer to associate with the Utp6/Utp18 dimer and that Utp18 is required for stoichiometric association of the heterotetramer with Utp6.

Taken together, these data can only be rationalized if Utp21 is indeed present at the foot structure. Its bent and elongated shape is further in agreement with the shape of the EM density map in this region.

3. In Fig. 6, experimental results gathered from static spatial interactions were extrapolated into a temporal binding model, with exciting biological implications. However, without reconstruction data of processome at various stages, the model can be quite inaccurate. The order of recruitment for various subcomplexes can be different from the linear sequence of the RNA.

We thank reviewer 3 for this comment. We agree that multiple scenarios are possible based on our data. In the revised manuscript we have added a sentence that addresses this point and highlights that we are only showing a model for one possible scenario:

“RNA-protein interactions identified by CRAC correspond to a population of temporal states. However, previous EM analyses of “Miller” chromatin spreads and analyses of metabolic labeling strongly indicate that early pre-ribosome assembly occurs co-transcriptionally (Gallagher et al 2004, Kos & Tollervey 2010). Therefore, while we cannot distinguish sequential RNA binding events of individual UtpA or UtpB subunits it seems likely that the 5' to 3' location of crosslinking sites will at least partially reflect the order of binding. In Figure 6 we present a model, for the potential, sequential RNA binding of UtpA and UtpB subcomplexes (Fig. 6).”

Reviewer #4 (Remarks to the Author):

In this manuscript, the authors used a wide spectrum of techniques to unveil the global architecture of two important protein complexes of the small subunit processome and provided insights into the functions of these biological complexes in the early stage of ribosome biogenesis. This study will pave the way for future structural characterization of the small subunit processome and further our understanding of the complex process of ribosome biogenesis in eukaryotes. The data are of high quality and the paper is well written.

We thank reviewer 4 for this positive assessment of our work.

However, there are several major points that need to be addressed to improve the investigation and to warrant publication in Nature Communications journal.

An obvious bit of missing data is the structure of the UtpA complex. Since the paper addresses the binding, complex interactions, etc., of both the UtpA and UtpB complexes, it is curious why the EM structure of purified UtpA complex was omitted from the study.

In fact, we did visualize the UtpA complex by negative-stain EM. However, while the complex isolated from yeast behaves well by size-exclusion chromatography, it has a very variable structure, which makes it impossible to calculate 3D reconstructions. We now include an additional supplementary figure that shows an EM image of negatively stained UtpA. In addition, we also show a gallery of particles that illustrates that UtpA consists of a globular core and an elongated arm domain that is flexibly tethered to the core and can adopt a wide range of orientations relative to the core.

There is also a question of stoichiometry within the two complexes. SDS-PAGE analysis of the purified complexes shows bands of varying intensities, which would suggest that the stoichiometry of the individual proteins is not necessarily equivalent.

We agree with reviewer 4 that in some cases the band intensities of subunits from the same complex are not identical. This is especially true for reconstituted subcomplexes of UtpB. The isolation of native complexes from yeast usually results in somewhat super-stoichiometric amounts of the tagged protein subunit, especially when stringent conditions are used, such as in Figure 1d where Utp10 and Utp17 were analyzed. In Figure 2, panels d and f, the band for Utp6 appears stronger as this was the tagged subunit when purifying Utp6/Utp18 from bacteria.

In the field, UtpA and UtpB are well known to form stoichiometric complexes based on previous publications (Krogan et al. Nature 2004 Fig. 3; Dosil et al. JBC 2004 Fig.7; Pöll et al. PLoS One 2014).

For UtpB, the only complex for which we have structural information, we have addressed this separately in a new supplementary Figure (Figure S3) in which the stoichiometric Utp12/Utp13 complex is analyzed by size-exclusion chromatography (panel a) and SDS-PAGE (panel b). Subsequently pull-down experiments show that, with the exception of Utp6, which was once again the tagged subunit, the other five subunits appear to be present in stoichiometric amounts when all subunits are immobilized (panel c).

Similarly, without knowing the stoichiometry and without density labels, the proposed placement of Utp21 in the EM volume is purely speculative. As such, the authors should refrain from docking of the x-ray structure without solid evidence of its location within the EM complex. Alternatively, the investigation would be more complete with density labels to unequivocally identify the localization of each protein within the complex.

The concern regarding the placement of Utp21 into the EM density map has been addressed in our response to reviewer 3 and the concern regarding the stoichiometry has been addressed in the response above.

Validation of DSS cross-linking and mass spectrometry analysis data is not included. For instance, the intra-molecule crosslinking observed in Utp21 or other parts with known structural information should be used to gauge the quality of the identified crosslinked peptides.

We agree with reviewer 4 that a structure-based validation of our DSS cross-linking data would be ideal and this has been done whenever possible in the past (Shi et al. Mol. Cell Proteomics 13 (2014) 2927-43). However, at this point, the only available crystal structure is that of a truncated version of Utp21, and there are no publicly available atomic structures for any of the other twelve protein subunits investigated in this paper. It is thus not possible to have a sense of the quality of the observed cross-links, in particular for the following reasons.

For almost all the intra-protein cross-links that we observe for Utp21, the responsible pairs of residues are not present in the crystal structure, which accounts for only 33 of the 63 lysine residues of Utp21.

These residues, which give rise to almost all cross-links, are either present in flexible loops, which were not built in the crystallographic model, or are present in the C-terminal region of Utp21.

Since we cannot independently validate the DSS cross-links in the absence of high-resolution structural information, we have not used DSS cross-links for detailed structural interpretation in this manuscript and have only used these to provide a general picture of protein-protein interactions within UtpA and UtpB. In the case of UtpB, for which we do have low-resolution structural information, we have independently verified direct protein-protein interactions by a series of biochemical experiments as shown in Figure 2 as well as the new Supplementary Figure 3.

Supplementary Figure 2 is a nice but coarse representation of the data. It will be useful if the author could include a quantitative score in the supplementary table for the observed crosslinked peptides. The current interpretation of DSS crosslinking data (inferring the proximity between two proteins based on the number of observed crosslinked peptides) is problematic. It is possible that more crosslinked peptides arise due merely to having more lysines around a particular protein interface. With a quantitative score, which tells the confidence level for the observed crosslinked peptide, the interpretation of the organization of protein subunits would be more convincing.

We agree that a quantitative score is useful to indicate a confidence level for the observed cross-linked peptides and in Supplementary Tables 3 and 4 we have now added an additional column with a quantitative score (calculated using pLink (Yang et al. 2012)).

We have further revised Supplementary Figure 2, which now indicates the distribution of lysines as well as intra- and inter-protein cross-links. We believe that this depiction is more intuitive and shows that lysine abundance is not the primary reason for our cross-links.

On pg. 9, the introduction to UV cross-linking requires greater detail. What does it mean to "target multiple subunits of each complex"? Instead of simply referring to previous publications for CRAC, more details should be provided for the application of this technique in this study. For example, how were the uv cross-linked protein-RNA complexes purified (denatured or native by tagging these proteins) and assigned to specific subunits? Since some protein bands on SDS-PAGE in Figure 1 and 2 were shown to be overlapped, it is not clear how the pattern would look like with crosslinked RNA.

We thank reviewer 4 for this comment and have added two sentences in the manuscript, which explain the general principles of CRAC for the broad readership of Nature Communications:

“A tripartite tag consisting of a polyhistidine tag, a TEV cleavage site, and a protein A tag (HTP) was inserted at the genomic loci encoding subunits of the UtpA and UtpB complexes, under the control of the endogenous promoters. Following in vivo UV cross-linking in actively growing cells, covalently bound RNA-protein complexes were subjected to multistep purification. Complexes were initially purified by binding to IgG Sepharose followed by TEV protease elution. Bound RNAs were then partially digested followed by two denaturing purification steps; nickel affinity purification in the presence of 4M guanidinium HCl and SDS-PAGE. These steps specifically select RNA that was covalently cross-linked to the tagged protein subunit, which was identified by RT-PCR and Illumina sequencing. Hence, only direct RNA-protein interactions are identified for a single protein subunit in a single experiment.”

Also, it is not clear that mutants are being used in the analysis, so the reference to "wild-type control" is lost. How does the wild-type complex serve as a negative control here? Again, more information regarding the CRAC analysis (within the results section) should help the reader.

The wild-type control refers to the wild-type yeast strain, which contains no tagged protein. To clarify this point further, we have changed the labels in Figures 4 and 5 to “non-tagged”.

Finally, the authors should elaborate on what is known about the "common contaminant" and include appropriate references to support the claims.

The “common contaminant” is a sequence within the 25S rRNA, which gives rise to peaks even in the absence of a tagged protein. This sequence has been observed in numerous previous publications, such as:

Granneman et al., EMBO 2010

Figure 1b

Schneider et al., Molecular Cell 2012 Figure 7a

Bradatsch et al. Nature NSMB 2012 Figure 4a

The above references have been inserted in the main text to mention this contaminant specifically.

On pg. 6 the authors mention that Utp12 and Utp13 co-elute on size-exclusion chromatography. However, the reference to figure 2c is a gel. Inclusion of the SEC profile (in supplement) is desired.

We agree with reviewer 4 and have included the SEC profile and corresponding gel in Supplementary Figure 3, panels a and b, respectively.

The inference of the orders of Utp subunits binding events is ambiguous. This information mainly comes from CRAC analysis, which probes these binding events under equilibrium state. The picture of whether these Utp complexes form first or binds to the transcripts then assemble is still not very clear.

In the revised manuscript we have added a sentence, which addresses this point (c.f. comment of reviewer 3 above). It is clear that UtpA and UtpB form complexes, which can be purified in the absence of nucleic acid, and which co-migrate on size-exclusion chromatography as shown in this manuscript.

The concluding paragraph/sentence in its current form ends abruptly. An expanded and more complete conclusion would improve the communication.

We agree that this last sentence was somewhat out of place and have deleted it so that the discussion has a different end.

Reviewers' Comments:

Reviewer #1 (Remarks to the Author)

Previously, it has been shown by the authors and others that early stages of ribosome assembly in yeast involve hierarchical assembly of protein complexes UtpA, UtpB, Mpp10 and the U3 snoRNP with precursor RNA destined for the small ribosomal subunit. This is a very nicely written manuscript describing analysis of the protein topology of the UtpA and B protein complexes and their interactions with rRNA as well as with the U3 snoRNA. The authors purify the native A and B complexes using affinity tags and gel exclusion chromatography. The A complex contains seven proteins while the B complex has six. Pairwise interactions between these proteins were discerned by salt washes, coexpressing protein pairs in yeast, and protein-protein cross linking. A low resolution structure of the B complex was obtained by EM, revealing an elongated structure with Utp12 and 13 at one end, Utp1 and 21 in the middle, and Utp18 and 6 at the other end, in a flexible module. Binding sites to pre-rRNA and U3 snoRNA were mapped by UV cross linking.

Together, these results led the authors to propose a working model for early stages of assembly that should be extremely useful for those studying ribosome biogenesis. UtpA binds to sequences at the 5' end of the 5'ETS followed by UtpB functioning as flexible, elongated bridge to recruit the U3 snoRNP and bring together the ends of pre-25S pre-rRNA.

The authors have done an excellent job to describe their work.

Did protein protein cross linking experiments reveal interactions between the UtpA and UtpB complex proteins?

Figure 2: What are the fainter bands seen in figure 2d,e,and f?

Figure 5: Is the 5' ETS really beige?

Reviewer #2 (Remarks to the Author)

The authors of this manuscript describe interaction studies and structural analysis of two subcomplexes, named UtpA and UtpB, which are involved in the assembly of ribosomes in yeast. The purification of the complexes is described, followed by identification of interactions between protein components by crosslinking analysis. The authors present a partial cryoEM structure of one of the complexes, UtpB and map the available crystal structure of one of the proteins into the densities. In addition, UV crosslinking is performed with several of the protein components of the UtpA and UtpB complexes and their binding sites on ribosomal RNAs and the snoRNA U3 are mapped.

The manuscript reports a large body of work that is presented well in the context of the literature. The main novelty of the work lies in the density map presented for the UtpB complex. However, the composition of the complexes as well as many of the interactions have been reported previously and the manuscript falls short of reporting the more detailed molecular interfaces and lacks functional aspects of the reported interactions. Unfortunately, hardly any structural information about the proteins that are part of the complex is available; only the structure of part of Utp21 could be fitted in the densities. In addition, the UV crosslinking data do not reveal novel information on the mechanistic of early ribosome assembly and this dataset is somewhat disjoint from the structural data and the information is not integrated beyond a very schematic model.

Without the addition of substantial datasets on the function or regulation of these interactions, the manuscript seems more suitable for a specialised journal.

Reviewer #3 (Remarks to the Author)

In this paper, the authors studied the molecular architecture of UtpA and UtpB complexes in ribosome small subunit processome, and their binding sites on pre-ribosomal RNA. The complexes and subcomplexes were characterized by multiple biochemical approaches, including gel-filtration chromatography, chemical cross-linking and mass spectrometric analysis, negative stain EM, UV cross-linking and sequencing. Consistent results on subunit interface mapping were generated from various approaches. Overall, the experiments were carefully designed and executed to produce biologically important results. The manuscript is suitable for publication on Nature Communications.

Here are a few minor points that the authors may consider in revision:

1. It seems that only Utp2, Utp1, Utp12 and Utp13 were included in chemical cross-linking study (page 6, 2nd paragraph). Is there a reason Utp18 and UtpG6 were not included? The UtpB complex seems stable enough for EM studies.
2. In EM density assignment (page 8), Utp21 was placed to the foot region, based on "biochemical and mass spectrometry experiments have shown that the tandem beta-propeller domain of Utp21 can interact with an N-terminal segment of Utp18". It is not clear which biochemical and MS experiments led to this conclusion.
3. In Fig. 6, experimental results gathered from static spatial interactions were extrapolated into a temporal binding model, with exciting biological implications. However, without reconstruction data of processome at various stages, the model can be quite inaccurate. The order of recruitment for various subcomplexes can be different from the linear sequence of the RNA.

Reviewer #4 (Remarks to the Author)

In this manuscript, the authors used a wide spectrum of techniques to unveil the global architecture of two important protein complexes of the small subunit processome and provided insights into the functions of these biological complexes in the early stage of ribosome biogenesis. This study will pave the way for future structural characterization of the small subunit processome and further our understanding of the complex process of ribosome biogenesis in eukaryotes. The data are of high quality and the paper is well written. However, there are several major points that need to be addressed to improve the investigation and to warrant publication in Nature Communications journal.

An obvious bit of missing data is the structure of the UtpA complex. Since the paper addresses the binding, complex interactions, etc., of both the UtpA and UtpB complexes, it is curious why the EM structure of purified UtpA complex was omitted from the study.

There is also a question of stoichiometry within the two complexes. SDS-PAGE analysis of the purified complexes shows bands of varying intensities, which would suggest that the stoichiometry of the individual proteins is not necessarily equivalent.

Similarly, without knowing the stoichiometry and without density labels, the proposed placement of Utp21 in the EM volume is purely speculative. As such, the authors should refrain from docking of the x-ray structure without solid evidence of its location within the EM complex. Alternatively, the investigation would be more complete with density labels to unequivocally identify the localization of each protein within the complex.

Validation of DSS cross-linking and mass spectrometry analysis data is not included. For instance, the

intra-molecule crosslinking observed in Utp21 or other parts with known structural information should be used to gauge the quality of the identified crosslinked peptides.

Supplementary Figure 2 is a nice but coarse representation of the data. It will be useful if the author could include a quantitative score in the supplementary table for the observed crosslinked peptides. The current interpretation of DSS crosslinking data (inferring the proximity between two proteins based on the number of observed crosslinked peptides) is problematic. It is possible that more crosslinked peptides arise due merely to having more lysines around a particular protein interface. With a quantitative score, which tells the confidence level for the observed crosslinked peptide, the interpretation of the organization of protein subunits would be more convincing.

On pg. 9, the introduction to UV cross-linking requires greater detail. What does it mean to "target multiple subunits of each complex"? Instead of simply referring to previous publications for CRAC, more details should be provided for the application of this technique in this study. For example, how were the uv cross-linked protein-RNA complexes purified (denatured or native by tagging these proteins) and assigned to specific subunits? Since some protein bands on SDS-PAGE in Figure 1 and 2 were shown to be overlapped, it is not clear how the pattern would look like with crosslinked RNA. Also, it is not clear that mutants are being used in the analysis, so the reference to "wild-type control" is lost. How does the wild-type complex serve as a negative control here? Again, more information regarding the CRAC analysis (within the results section) should help the reader. Finally, the authors should elaborate on what is known about the "common contaminant" and include appropriate references to support the claims.

On pg. 6 the authors mention that Utp12 and Utp13 co-elute on size-exclusion chromatography. However, the reference to figure 2c is a gel. Inclusion of the SEC profile (in supplement) is desired.

The inference of the orders of Utp subunits binding events is ambiguous. This information mainly comes from CRAC analysis, which probes these binding events under equilibrium state. The picture of whether these Utp complexes form first or binds to the transcripts then assemble is still not very clear.

The concluding paragraph/sentence in its current form ends abruptly. An expanded and more complete conclusion would improve the communication.